

# An association between feather damaging behavior and corticosterone metabolite excretion in captive African grey parrots (*Psittacus erithacus*)

Pierluca Costa[1,*], Elisabetta Macchi[1,*], Emanuela Valle[1,*], Michele De Marco[1], Daniele M. Nucera[2], Laura Gasco[2] and Achille Schiavone[1]

[1] Department of Veterinary Science, University of Turin, Grugliasco (TO), Italy
[2] Department of Agricultural, Forest and Food Sciences, University of Turin, Grugliasco (TO), Italy
[*] These authors contributed equally to this work.

Corresponding authors
Pierluca Costa,
pierluca.costa@unito.it
Elisabetta Macchi,
elisabetta.macchi@unito.it
Emanuela Valle,
emanuela.valle@unito.it
Achille Schiavone,
achille.schiavone@unito.it

## ABSTRACT

**Background**. African grey parrots (*Psittacus erithacus*) are kept as pets and are frequently hand-reared. It has been observed that hand-reared African grey parrots may develop behavioral disorders such as feather damaging behavior (FDB). It is well known that chronic stress is involved in behavioral disorders in captive parrots. The main glucocorticoid in birds is corticosterone; its quantification provides information about adrenocortical activity and is considered to be a reliable indicator of stress levels in birds. We analyzed the differences in the excretion of corticosterone metabolites (CM) in the droppings of African grey parrots characterized by: 1. different rearing histories (parent rearing vs. hand rearing); and 2. the presence or absence of FDB in hand-reared parrots.

**Methods**. A total of 82 African grey parrots that were kept in captivity were considered. According to breeding methods, three groups of birds were defined: 1. The parent-reared (PR) parrots included birds kept in pairs ($n = 30$ pairs) with a conspecific partner of the opposite sex. All of these birds were healthy and never showed FDB signs; 2. The healthy hand-reared parrots (H-HR) included pet parrots individually kept, that were hand-reared and did not display any sign of FDB ($n = 11$, 7 males and 4 females); 3. The FDB hand-reared parrot (FDB-HR) included pet parrots individually kept, that were hand-reared and displayed FDB ($n = 11$, 7 males and 4 females). Droppings were collected in the morning over three alternating days in autumn 2014 and spring 2015. The CM were determined using a multi-species corticosterone enzyme immunoassay kit. Split-plot repeated-measure ANOVA was used to examine any differences using group, season and group × season as the main factors.

**Results**. Different quantities of CM in droppings were found for the three groups. The mean CM value was 587 ng/g in the PR parrots, 494 ng/g in the H-HR parrots and 1,744 ng/g in the FDB-HR parrots, irrespective of the season. The excretion of CM in FDB-HR was significantly higher than in PR or H-HR parrots. CM in droppings were not influenced by the season (autumn vs. spring); furthermore, the interaction between group and sampling season was not significant. Limited to the H-HR and FDB-HR groups, a trend in the significance of the difference in the mean CM excreted by male and female birds was observed, with the levels excreted by males being higher than

those excreted by females. When the effect of age was considered (in the two separate groups), there was a statistically significant positive correlation only for H-HR. **Conclusions**. The highest amount of CM excretion was found in FDB-HR parrots, and a positive correlation between age and CM excretion was found in H-HR. Given that the CM excretion of both PR and H-HR parrots was similar in our study, future research is recommended to investigate the specific aspects of hand-rearing to improve parrot welfare.

## INTRODUCTION

African grey parrots (*Psittacus erithacus*) are kept as pets in private households because of their sociability and also for their ability to imitate human speech.

African grey parrots may be hand-reared, and this practice has been increasingly carried out over the last 30 years. Based on the hand-rearing method used, hand-reared parrots can be divided into different groups according to the incubation system (natural vs. artificial) and the age of removal from the nest (at hatch, less than approximately five weeks or more than approximately five weeks) (*Schmid, Doherr & Steiger, 2006*). In contrast to parent-reared parrots, which imprint toward conspecifics (*Glendell, 2003*), hand-reared parrots imprint on humans and seem to be socially dependent on them. The exact consequences of the different hand-rearing methods on the development of behavior in adult birds are still not clear. However, it has been observed that hand-reared grey parrots may develop behavioral disorders, such as aggressiveness, feather picking, stereotypies or abnormal sexual behaviors, and thus it is expected that they are prone to develop such behavioral disorders (*Schmid, Doherr & Steiger, 2006*). Moreover, is has been observed that hand-reared chicks that were less than 5 weeks old when removed from the nest developed stereotypies more often than chicks that stayed longer with their parents (*Schmid, Doherr & Steiger, 2006*).

Feather damaging behavior (FDB) includes plucking, chewing, fraying and/or biting, and it results in the loss of or damage to feathers (*Van Zeeland et al., 2009*; *Van Zeeland et al., 2013*). FDB in parrots is usually self-inflicted and generally includes all mutilation of the feathers accessible to the bird's beak (*Harrison, 1986*). *Grindlinger (1991)* estimated that approximately 10% of the captive bird population suffered from FDB. *Kinkaid et al. (2013)*, in a sample of 538 parrots, found an FDB prevalence of 15.8%. Our group previously conducted a study considering this classification, which showed a notable difference in the FDB prevalence in the two different populations of parrots. The parent-raised population ($n = 1{,}488$) showed an FDB prevalence of 1.3%, while the pet parrot population ($n = 292$) showed an FDB prevalence of 17.5% (*Costa et al., 2016*). FDB has rarely been observed in the wild and usually occurs in captive birds when they reach sexual maturity (*Wedel, 1999*), even though some authors have reported the onset of FDB prior to the occurrence of sexual maturity (*Jayson, Williams & Wood, 2014*). FDB occurs in many species of parrots, and it has been observed in African grey parrots (*Psittacus erithacus*) and

cockatoos (*Cacatua* spp.) (*Clubb et al., 2007*; *Jayson, Williams & Wood, 2014*; *Peng et al., 2014*), *Amazona* spp. parrots (*Garner et al., 2006*), *Ara* spp. and *Agapornis* spp. (*Costa et al., 2016*), crimson-bellied conures (*Pyrrhura perlata*, *Van Hoek & Ten Cate, 1998*) and other psittacine species. It has been suggested that FDB could be a coping strategy for negative affective states (e.g., stress and boredom) and/or living in a suboptimal environment (*Rosskopf Jr & Woerpel, 1996*; *Levine & Practice, 2003*). In many cases, these patterns may represent an exaggeration or expansion of normal behavior, resulting from inadequate environmental stimuli and/or early weaning and/or social isolation (*Garner et al., 2006*; *Schmid, Doherr & Steiger, 2006*; *Van Zeeland et al., 2013*).

It is well known that chronic stress is involved in behavioral disorders in captive parrots (*Ferreira et al., 2015*; *Owen & Lane, 2006*). In vertebrates, the front-line hormones for overcoming stressful situations are $\beta$-endorphin, glucocorticoids and catecholamines (*Ayala et al., 2012*; *Johnstone, Reina & Lill, 2012*; *Livingston, 2010*; *Möstl, Rettenbacher & Palme, 2005*; *Schmidt et al., 2010*). The main glucocorticoid in birds is corticosterone; its quantification provides information about adrenocortical activity (*Ferreira et al., 2015*) and is considered to be a reliable indicator of stress levels in birds (*Dehnhard et al., 2003*; *Hartup et al., 2004*; *Young & Hallford, 2013*), giving important insight into the welfare status of an individual or a group of animals (*Lane, 2006*), especially when used in conjunction with other parameters such as behavior. The analysis of fecal corticosterone is preferred over blood sampling because is less invasive and can cause fewer stress responses (*Nemeth et al., 2016*) without compromising the welfare assessment (*Hamilton & Weeks Jr, 1985*; *Le Maho et al., 1992*). Several authors have reported a correlation between the concentrations of plasma glucocorticoids and their metabolites in the feces of mammals (*Möstl et al., 1999*; *Palme et al., 1999*; *Stead, Meltzer & Palme, 2000*, *Palme et al., 2005*) or in the droppings of birds (*Dehnhard et al., 2003*).

*Owen & Lane (2006)* measured corticosterone in the droppings of African grey parrots, and they observed that the corticosterone level in the excreta of FDB parrots was higher than that of healthy pet parrots. However, these authors did not consider the sex and age of the parrots or the season in which the samples were taken. The purpose of the present study was to compare the excretion of corticosterone metabolites (CM) in the droppings of hand-reared (with or without FDB) and in parent-reared African grey parrots (kept in pairs for reproduction) during autumn and spring. For the hand-reared parrots, the influence of sex and age on the amount of corticosterone in droppings was also considered. An increase in CM in hand-reared parrots with FDB was expected. Furthermore, we aimed to determine if healthy hand-reared parrots and parent-reared parrots display similar patterns in CM excretion.

## MATERIALS & METHODS

### Animal and selection criteria

The study was based on a web questionnaire used in a previous study (*Costa et al., 2016*) that was addressed to the owners of all species of pet parrots. The questionnaire was distributed throughout Italy through online parrot association sites, social networks and e-mails in collaboration with the Italian Psittacine Club (known as the "Club degli Psittacidi"

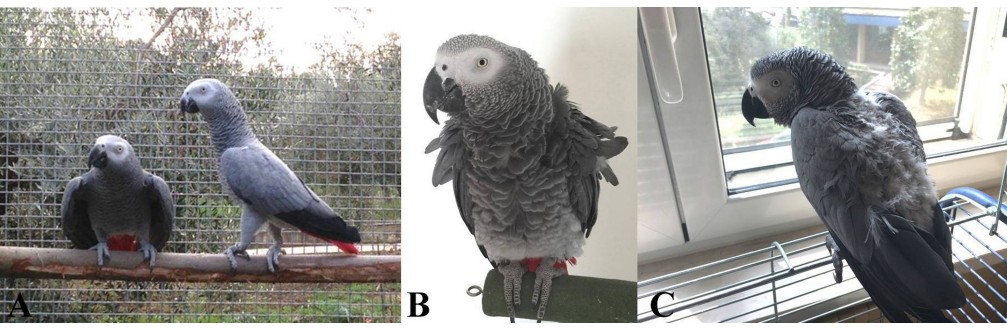

**Figure 1** **African grey parrots (*Psittacus erithacus*) observed in the present study.** (A) A parent-reared pair; (B) healthy hand-reared parrot and (C) hand-reared parrots that display feather damaging behavior.

http://psittacidi.webservice-4u.com/) and the Italian Association of Parrot Breeders (known as the "Associazione Italiana Allevatori Pappagalli," http://www.assopappagalli.it/). In the present study, we only considered African grey parrots because this was the most represented species among the Italian respondents (*Costa et al., 2016*) and because this species is considered to be very sensitive to FDB (*Jayson, Williams & Wood, 2014*; *Schmid, Doherr & Steiger, 2006*).

All birds considered in our study were born in captivity, and no wild-caught birds were used. A total of 82 African grey parrots were considered. To be included in the study, the birds had to be at least thirty-six months old, so that only birds that had a fully formed character and sexual behavioral patterns were considered. Based on the different methodologies of rearing at the neonatal stage, hand-reared and parent-reared parrots were considered. Among the hand-reared parrots, a further distinction was made between parrots displaying FDB and parrots not displaying FDB. According to these criteria, three samples of birds were defined: 1. Parent-reared (PR) parrots; 2. Healthy hand-reared parrots (H-HR); 3. FDB hand-reared parrots (FDB-HR).

1. The parent-reared (PR) parrots (Fig. 1A) included birds kept in pairs ($n = 30$ pairs) with a conspecific partner of the opposite sex, since they were specifically reared for reproduction. These birds were reared by their biological parents, and contact with humans was minimal and related only to their care and daily management. The PR parrots were permanently housed in a standard parrot cage with a minimum volume of 1 m$^3$ and exposed to natural light variation. All of these birds were healthy and never showed signs of FDB. All of the birds included in this sample were housed in the same facility. We included this sample that we considered a valid control for stress coping since (usually considered well-balanced birds that have learnt all of the specific behavioral patterns of their species). We included this sample since parent-reared captive parrots are usually considered well-balanced birds that have learnt all of the specific behavioral patterns of their species (*Schmid, Doherr & Steiger, 2006*).

2. The healthy hand-reared parrots (H-HR) (Fig. 1B) included pet parrots that were hand-reared. These birds did not display any sign of FDB. This sample was composed of 11 birds (7 males and 4 females). Each bird was individually kept by a owner.
3. The FDB hand-reared parrots (FDB-HR) (Fig. 1C) included pet parrots that were hand-reared. These birds displayed FDB. This sample was composed of 11 birds (7 males and 4 females). Each bird was individually kept by a owner. The diagnosis of FDB was made by a veterinary expert in exotic birds who took into consideration all of the possible differential diagnoses according to *Van Zeeland et al. (2009)*. In this way, it was possible to rule out any clinical problems.

The H-HR were age (±2 years) and sex matched with the FDB-HR. Both H-HR and FDB-HR parrots lived mostly outside a cage without any other parrots and had a close relationship with humans. All of the birds were privately owned and had free access to water and to commercial diets formulated specifically for parrots that were supplemented with fruit and vegetables. The owners of all of the parrots included in the study completed a questionnaire about the care and management of the parrots and, only for FDB-HR parrots, the main body regions affected by FDB.

## Droppings sampling and analysis

Droppings were collected throughout autumn 2014 and spring 2015 in the middle of each season. The droppings were collected in the morning (9:00–11:00 AM) for three days on alternating days. This time frame was chosen with the intention to reduce the effect of daily patterns in CM excretion. The samples were collected directly from the cleaned bottom of the bird's habitual cage where the parrot lived. For PR parrots, the dropping samples represent a pool of the excreta from the parrot pairs, whereas the droppings were individually collected for the H-HR and FDB-HR parrots. The 3-day samples were pooled, stored in 50-mL plastic tubes and immediately frozen at −20 °C until analysis. A total of 30, 11 and 11 samples were collected at each sampling time for the PR, H-HR and FDB-HR parrots, respectively.

To extract steroids, we used the methanol-based procedure described by *Palme et al. (2013)* with slight modifications. Briefly, the droppings were lyophilized, weighed, and completely crushed, and two aliquots of the samples (0.25 g each) were placed into extraction tubes, which were then sealed with a Teflon cap and stored at −20 °C. Each aliquot was thoroughly mixed for 30 min using a multivortex with one mL of 80% methanol (Sigma Aldrich, St. Louis, MO, USA). The suspension was then centrifuged at 500 g for 20 min and the supernatant was recovered. An aliquot (0.5 mL) of the supernatant was transferred into a new vial and evaporated at 50 °C for 14 h. After evaporation, the dried extracts were stored at room temperature in dark boxes for 15 days and then kept at −80 °C until they were assayed. One day before the CM analyses, the dried extracts were re-diluted in 0.5 mL of 80% methanol. An aliquot of the extract was diluted to 1:10 in the assay buffer (Arbor Assays®, Ann Arbor, MI, USA). The mixture was then vortexed and left to rest for 5 min twice to ensure complete steroid solubility. The CM were determined using a multi-species corticosterone enzyme immunoassay kit (K014; Arbor Assays®, Ann Arbor, MI, USA). All of the analyses were repeated twice. The inter- and intra-assay coefficients of variation were less than 10% (6% and 8%, respectively). The sensitivity of the assay was 11.2 ng/g droppings. All of the droppings samples were analyzed at multiple dilutions (1:4, 1:8, 1:16 and 1:32), and all regression slopes were parallel to the standard curve ($r^2 = 0.983$).
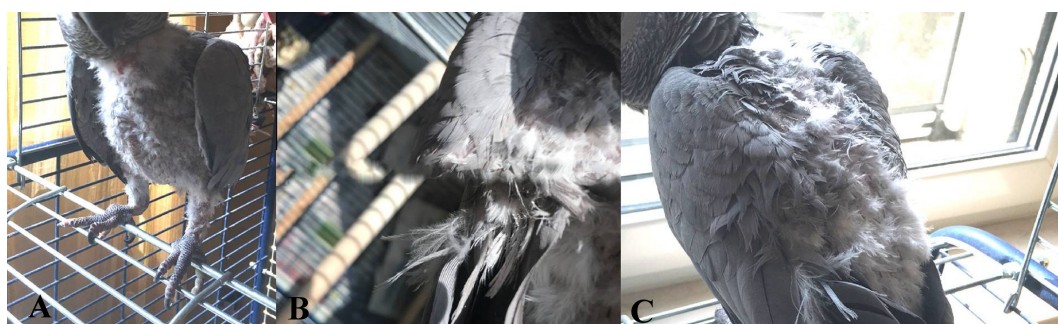

**Figure 2** **Deplumation area in feather damaging behavior African grey parrots.** (A) Chest area; (B) wings; (C) shoulders and rump.

The mean recovery rate of corticosterone added to dried excreta was 95.8%. According to the manufacturer, the corticosterone kit presents the following cross reactivity: 100% with corticosterone, 12.3% with desoxycorticosterone, 0.62% with aldosterone, 0.38% with cortisol and 0.24% with progesterone. The concentration of CM was expressed as ng/g of droppings dry matter.

## Data analysis

The CM of the PR, HP and FDB-P parrots were compared. Before testing for group differences, the normality of the data distribution and the homogeneity of variance were assessed using the Shapiro–Wilk test and Levene's test, respectively. Split-plot repeated-measure ANOVA was used to examine any differences using one within-subject variable (season) and one between-subject variable (the three samples of birds) and considering the interaction between these main effects. When the main effect was significant, a Tukey's post hoc test was performed to analyze the differences between groups. To explore the effects of sex and age on CM within the H-HR and FDB-HR groups, a $t$-test and a correlation analysis (Pearson's $r$) were performed, respectively. The data are presented as the mean and the pooled standard error of the mean (SEM). Statistical significance was set at 0.05, and a trend of significance was considered at $p < 0.1$. All statistical analyses were performed using SPSS version 15.0 for Windows (SPSS Inc., Chicago, IL, USA).

## RESULTS

The average age of the birds was $8.1 \pm 1.7$, $7.9 \pm 5.4$, $7.8 \pm 5.4$ years for PR, H-HR and FDB-HR parrots respectively. The average volume of the aviary cages in which the birds belonging to the PR group were kept was 4.85 m$^3$. The average volume of the cages of each H-HR and FDB-HR parrot was 1.70 m$^3$, although they were kept outside the cage on a daily basis for at least five hours, thus living in close contact with their owners. The main region affected by FDB in the FDB-HR birds was the chest (90.9%) (Fig. 2A), and this was followed by the wings (18.2%) (Fig. 2B), the shoulders and the rump (9.1%) (Fig. 2C). No sign of FDB was observed on the head.

Different quantities of CM in droppings were found for the three samples of African grey parrots. The mean CM value was 587 ng/g in the PR parrots, 494 ng/g in the H-HR

**Table 1** Corticosterone metabolite (ng/g dry matter) excretion in the droppings of healthy and FDB[a] African grey parrots (*Psittacus erithacus*) (mean and pooled SEM).

| Group | Season | Mean | SEM | 95% confidence interval | |
|---|---|---|---|---|---|
| | | | | Lower bound | Upper bound |
| PR[b] parrots | Autumn | 617 | 25 | 558 | 676 |
| | Spring | 558 | 31 | 467 | 649 |
| | Mean | 587 | 20 | | |
| H-HR[c] parrots | Autumn | 519 | 58 | 421 | 616 |
| | Spring | 469 | 45 | 318 | 620 |
| | Mean | 494 | 36 | | |
| FDB-HR[d] parrots | Autumn | 1,749 | 55 | 1,652 | 1,847 |
| | Spring | 1,739 | 133 | 1,589 | 1,890 |
| | Mean | 1,744 | 70 | | |
| Main effects: | | | | | |
| Group: | | $F = 194.477$ | $p < 0.0001$ | | |
| Season: | | $F = 1.305$ | $p = 0.259$ | | |
| Interaction (group × season): | | $F = 0.191$ | $p = 0.826$ | | |
| Group contrasts (LSD test): | | | | | |
| PR vs. H-HR parrots: | | $p = 0.140$ | | | |
| PR vs. FDB-HR parrots: | | $p < 0.0001$ | | | |
| HR vs. FDB-HR parrots: | | $p < 0.0001$ | | | |

Notes.
[a] FDB, feather damaging behavior.
[b] PR, parent-reared.
[c] H-HR, healthy hand-reared.
[d] FDB-HR, feather damaging behavior hand-reared.

parrots and 1,744 ng/g in the FDB-HR parrots, irrespective of the season (Table 1). The excretion of CM in FDB-HR parrots was higher than in PR and H-HR parrots ($p < 0.001$). CM in droppings were not influenced by the season (autumn vs. spring); furthermore, the interaction between parrot groups and the sampling season was not significant (Table 1).

To explore the effect of sex on CM excretion in the H-HR and FDB-HR samples, a *t*-test was performed, considering the mean CM amount (autumn and spring) for each bird, given the non-significance of the within-subject effect (sampling season); moreover, in these samples, a correlation analysis (Pearson's *r*) using the same response variable was conducted to assess the effect of age on CM excretion. The results showed that there was a trend in the difference in the mean CM excreted by male and female birds, with the levels of males being higher than those presented in females: HP, mean of males = 554, mean of females = 388 ($t = 1.851$, $p = 0.097$); FDB-HR, mean of males = 1,852, mean of females = 1,556 ($t = 1.906$, $p = 0.089$). When the effect of age was considered (in the two separate populations), there was a statistically significant positive correlation only for H-HR ($r = 0.609$, $p = 0.047$); in contrast, no correlation was found for FDB-HR ($r = 0.398$, $p = 0.225$).

## DISCUSSION

In our study, we observed increased excretion of CM in FDB-HR parrots, which was approximately three times higher than that of PR and H-HR parrots, irrespective of the season of sampling. Moreover, no differences were found in CM excretion between H-HR and PR parrots, the latter of which were kept in pairs for reproduction and so they can maintain sexual and social activity.

Our results confirm the findings of *Owen & Lane (2006)*, which showed higher CM in the droppings of FDB-HR parrots than in control parrots. To the best of our knowledge, the paper of *Owen & Lane (2006)* is the only study comparing the CM excretion in droppings of FDB and non-FDB African grey parrots (261 ng/g and 75 ng/g, respectively). Our results confirm these observations in terms of significant differences between FDB-HR and H-HR parrots, but the magnitude of the values measured in our study was more than six times higher than those observed by *Owen & Lane (2006)*. In the study of *Owen & Lane (2006)*, the control group was composed of ten birds that were kept all together in a large aviary, so they presumably maintained their social and sexual activity. In contrast, in our study, we considered two samples of parrots that did not display FDB: PR and H-HR, which both display similar levels of CM excretion. Parent reared parrots are usually considered well balanced birds since parent rearing methods let them to learn all the specific behavior pattern, which is a great benefit for their welfare (*Schmid, Doherr & Steiger, 2006*). The link between FDB and the corticosterone levels of excreta has also been observed by *Peng et al. (2014)* in two cases of FDB in sulphur-crested cockatoos (*Cacatua galerita*); the authors found a decrease in corticosterone levels after treatments that consisted of socialization, a training program, medication and feeding enrichments. Even though we did not measure the environmental or enrichment management and the activity of parrots included in our study, it has been previously demonstrated that parrots with FDB display higher activity compared to parrots without FDB in a number of behavioral tests, suggesting that FDB is a proactive stress response pattern; under chronic stress conditions, proactive birds seemed to be more prone to develop behavioral disorders (*Van Zeeland et al., 2013*). FDB can lead to, or result from, underlying skin pathologies that itch or irritate (*Garner et al., 2008*). FDB may also cause health problems related to tissue damage, hemorrhage, infection, or hypothermia (*Meehan, Millam & Mench, 2003*; *Van Zeeland et al., 2009*). In the present study, the body area most affected by FDB was the chest, and the head was not affected by FDB. The presence of feathers in good condition in areas of the body that are not directly reachable (i.e., the head) by the birds is one of the criteria that has been used to make a distinction between FDB and other skin or plumage diseases (*Galvin, 1983*; *Harrison, 1986*; *Westerhof & Lumeij, 1987*).

The higher CM excretion in the FDB-HR parrots than in the H-HR and PR parrots, suggest an increase in adrenal cortical activity (*Möstl & Palme, 2002*). The adrenal glands have a key role in the hormonal response to short-term and chronic stress, which result in an increase in glucocorticoid secretion (*Möstl & Palme, 2002*). The measurement of CM in bird droppings has been proposed to assess the welfare status of birds (*Meehan, Garner & Mench, 2004*; *Van Zeeland et al., 2009*; *Cussen & Mench, 2015*; *Young & Hallford,*

*2013*; *Ferreira et al., 2015*; *Shepherdson, Carlstead & Wielebnowski, 2004*), the results of such analyses are hard to interpret because the biological perspective suggests only an increase in adrenocortical activity. For these reasons, the importance of these data could lead to misinterpretation because they are a result of a complex interaction between a wide range of physiological, endocrine and behavioral variables that occur simultaneously (*Gaskins & Bergman, 2011*; *Cussen & Mench, 2015*; *Van Zeeland et al., 2009*).

In our sample of hand-reared parrots, a trend in the significance of the difference in the mean CM excreted between male and female birds was found for both H-HR and FDB-HR parrots, with the levels excreted by males being higher than those excreted by females. In contrast, *Ferreira et al. (2015)* did not find any gender effect in the CM excretion of blue-fronted parrots (*Amazona aestiva*). Furthermore, a positive correlation between age and CM excretion was found for H-HR parrots. However, these results should be considered with caution in both studies due to the small sample sizes and the different species considered. The demographic features of FDB (i.e., sexual maturation) and gender predisposition (female > male) have been reviewed by *Van Zeeland et al. (2009)*, who state that the literature on this topic is related to field studies of small group of animals and that consequently larger surveys are thus necessary to confirm these results.

FDB is observed mainly in hand-reared parrots, occurring in from 10 to 17.5% of individuals (*Grindlinger, 1991*; *Kinkaid et al., 2013*; *Costa et al., 2016*), while in parent-reared parrots, FDB does not occur or occurs rarely (approximately 1%) (*Costa et al., 2016*). Hand-rearing has been considered to be a risk factor in the incidence of FDB (*Costa et al., 2016*; *Schmid, Doherr & Steiger, 2006*). Furthermore, social isolation and sexual behavior frustration can have important roles in the development of abnormal behavior (*Lantermann, 1989*; *Harrison, 1994*; *Van Hoek & Ten Cate, 1998*; *Wedel, 1999*; *Fox, 2006*; *Jayson, Williams & Wood, 2014*). According to *Fox (2006)*, abnormal sexual imprinting and a strong social preference for humans may cause behavioral problems in pet parrots, which are most likely more prone to inappropriately direct sexual behavior toward their owners. Since both H-HR and FDB-HR were in social and reproductive isolation in our study, this suggests that there is something different about their management that could be linked to environmental enrichment or breeding methods; thus, from an animal welfare perspective, it is fundamental to deeply research the risk factors that are involved in the incidence of FDB.

## CONCLUSIONS

In the present study, we analyzed the differences in CM excretion between African grey parrots characterized by: 1. different rearing histories (parent rearing vs. hand rearing); and 2. the presence or absence of FDB in hand-reared parrots.

The highest amount of CM excretion was found in FDB-HR parrots, and a positive correlation between age and CM excretion was found in H-HR.

Given that the CM excretion of both PR and H-HR parrots was similar in our study, future research is recommended to focus on the specific aspects of hand-rearing needed to improve the welfare of parrots.

## ACKNOWLEDGEMENTS

The authors would like to thank the parrot owners for providing samples and for filling out the questionnaire for the data collection effort as well as the various veterinary surgeons for diagnostic confirmations. The authors are also grateful to Dr. Valentina Ballabio, Miss Federica Ardizzone and Mr. Simone Durigon for the support provided during the organization of the research.

### Funding

This research was supported by the University of Turin (Italy)—Local Researc Program (ex 60%) 2014. The funders had no role in study design, data collection and analysis, decision to publish, or preparation of the manuscript.

### Grant Disclosures

The following grant information was disclosed by the authors:
University of Turin.

### Competing Interests

The authors declare there are no competing interests.

### Author Contributions

- Pierluca Costa conceived and designed the experiments, performed the experiments, wrote the paper, prepared figures and/or tables.
- Elisabetta Macchi conceived and designed the experiments, performed the experiments, wrote the paper.
- Emanuela Valle analyzed the data, wrote the paper, prepared figures and/or tables.
- Michele De Marco prepared figures and/or tables, reviewed drafts of the paper.
- Daniele M. Nucera analyzed the data.
- Laura Gasco contributed reagents/materials/analysis tools, reviewed drafts of the paper.
- Achille Schiavone conceived and designed the experiments, analyzed the data, contributed reagents/materials/analysis tools, wrote the paper, prepared figures and/or tables.

### Data Availability

The raw data has been supplied as Data S1.

### Supplemental Information

Supplemental information for this article can be found online at http://dx.doi.org/10.7717/peerj.2462#supplemental-information.

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
