# Peer review of "An association between feather damaging behavior and corticosterone metabolite excretion in captive African grey parrots (Psittacus erithacus)"

_PeerJ, doi:10.7717/peerj.2462_

## Round 0.1 · original submission · Major Revisions

I consider that the three reviewers of this manuscript have done an excellent job in identifying flaws and difficulties that must be resolved before the work could be accepted for publication.

I fully agree with Reviewer 2 that a repeated measures analysis is required - this reviewer has gone beyond the normal call of duty in re-analysing the data herself. It is reassuring that the main group effect still stands, but there are other nuances that will required some re-writing. All reviewers request more clarity around the methods (e.g. how were samples collected on alternate days at each season when there is only one sample per season. It is misleading to claim that there are 82 birds in the study, since 60 of them were in pairs where, it appears, individual samples could not be obtained. The reviewers also highlight uni-directional reasoning (stress causes feather picking) but other possible causal relationships are possible (e.g. feather picking arises from an unknown cause but pain or damage due to feather loss then causes corticosterone increase). Reviewer 1 considers some of the reasoning to be circular since the assay itself has not been validated for these subjects, so please address these points. The literature surrounding the relationship between stress response and stereotypy in birds is far more extensive and nuanced than you have acknowledged. Suggestions have been given for additional reading to improve the re-submission and, especially the research question and hypothesis. Finally, please consider moderating your conclusions to take account of potential (unknown) variations in bird management in the different groups, confounding factors and methodological constraints.

·

Basic reporting

see below

Experimental design

see below

Validity of the findings

see below

Additional comments

The paper of Costa et al aims at finding differences in adrenocortical activity (assessed non-invasively by measuring fecal corticosterone metabolites - FCM) in African grey parrots (both sexes) differently reared, and especially between those exhibiting feather pecking and the other groups. The authors report higher FCM levels in feather picking birds, irrespective if animals were in or out of the breeding season.
Unfortunately my impression is that the paper suffers from a serious lack of insight into both methodological and physiological issues around a determination of glucocorticoid metabolites in the feces (sorry for having to say this). I also think the knowledge gap being investigated is not clearly identified, and statements to how the study contributes to filling that gap are missing or respective conclusions not based on the data. Another drawback of the study is the low number of samples/individual analysed (in combination with mixed sex samples in WT animals).

General comments:
In most bird species (and I think parrots are no exception) urine and feces are released together in the form of droppings. Thus measured metabolites are not exclusively “fecal” metabolites.
“Corticoids”: Although I know that this term is frequently (mis-)used, I suggest to be precise: Both mineralocorticoids (aldosterone) and glucocorticoids may be labelled with this term. As only the latter are meant here, I suggest using “glucocorticoids” instead throughout the manuscript. Besides: The authors label the substances measured with their immunoassay “fecal corticosterone” – (e.g., title or abstract). However, they give no indication what they really measured (no HPLC was performed to characterize metabolites). There is ample evidence from the literature that glucocorticoids from the blood are heavily metabolized and thus not present in the feces of animals (in birds radioinfusion experiments demonstrate this – e.g. Rettenbacher et al. 2004). Therefore the term “fecal corticosterone” should be avoided, and “fecal corticosterone metabolites” used throughout the manuscript (to make it more readable an abbreviation such as FCM may be used).
I think relevant prior literature is not appropriately referenced. That’s especially the case for FCM analysis, but I also miss at least a short statement about FP in chicken which is a huge problem there and thus a big research topic. In addition, several important papers dealing with FCM and parrot species are not even mentioned (Dahlin et al., 2014; Ferreira et al., 2015; Young et al., 2013). As the authors seem to lack profound knowledge in the field of fecal steroid analysis they are encouraged to dig deeper into the relevant literature (e.g., review by Touma & Palme 2005; Goymann, 2005; but also other ones – see further references given at the end).
Most critically, there seems to be a deep misunderstanding about validation (unfortunately the authors may have been mislead by the manufacturer of the commercial EIA kit): You cannot validate an assay for a certain sample material (here: “dried fecal extracts” – by the way: if extracts are dried down you cannot analyse them)! There’s ample literature available and it is state of the art in non-invasive monitoring of stress hormones that an assay has to be validated for each species and sample material. In the case of fecal samples it is mandatory to physiologically (or biologically) validate an assay for a given species. This is achieved by demonstrating that an increase in adrenocortical activity (classically provoked by an ACTH challenge test) is reflected by an increase in FCM concentrations (found after a certain delay time). For a good example of such a validation in another parrot species please see Ferreira et al. (2015).
Important point: (see lines 272 onwards): I see the following main problem with the paper: You need a validated method to be able to draw sound conclusions in experiments (here animal welfare related ones). Or you can also use a known stressful situation (FP) to biologically validate your FCM method (and based on expected higher levels in FPP birds it seems arguable). But you cannot have both at once (that’s a sort of a circular reasoning). Hope I could make this important point comprehensible.
In addition, the manuscript needs improvement in grammar and style.

Further specific comments (listed by appearance in the manuscript):

Line 22: What is “ethological distress”?
Line 34 (+162): “pan-specific”: I have never before encountered this term to describe an assay: What does it mean?
Lines 39-40 (and elsewhere): Giving decimal points for FCM concentrations doesn’t make much sense (especially given the fact that repeatability of the assays is about 10%).
Line 43: The results don’t suggest 1). The authors found that FP animals excreted highest concentrations of FCM. The actual conclusion (stress plays a role in FP) is very vague and in the same way true before (or without) this experiment (on the basis of the cited literature). No clue if it is a cause or consequence?
Lines 110-112: That sounds very odd (due to gramatical problems).
Line 131: I question that all birds received the same standard commercial diet, because they are privately owned, or?
Lines 138-143: I’m somewhat confused about breeding seasons. Why is the breeding season shifted? And if this is possible there might still be an effect of the inherent natural breeding season superimposed by the actual breeding and activation of the reproduction. And there may also be sex effects found in FCM levels in the WT birds (which might have been masked by taking pool samples of the pairs – if I conclude this correctly from the data-file; if so it should also be clearly stated in the respective M&M section).
Lines 143-145: So you collected pool samples from individual birds (or couples) – and from how many days (“alternate”?) – at what months during the different seasons (the actual month may play a role – start/middle/end of season – especially when animals are not bred according to their natural habits)
Lines 152-154: In case of birds, pure ethanol is not the best option for extracting FCM (though again this is unfortunately recommended by the manufacturer). In the droppings fecal and urinary metabolites are combined and thus a high proportion of FCM is conjugated, resulting in large amounts of relatively polar metabolites. Therefore a lower percentage of an alcohol is recommended for extraction in birds (Palme et al., 2013).
Line 164: Give sensitivity of the total method (in ng/g droppings).
Lines 167-168: How can fecal samples be parallel to the standard curve?
Line 195: Fecal corticosterone concentrations were not influenced …
Lines 205-206: Results should not be first mentioned in the discussion.
Lines 222-223: Which way: higher or lower values?
Lines 237-241: Unfortunately Peng et al. (2014) also lacks a sound method for FCM analysis (Besides, the interpretation of the results given in Figs 2+3 seems questionable).
Lines 267-268: As outlined before (no physiological validation of the method provided!), I don’t think that the conclusion is based on the data.

References (cited above) which should be helpful:
Dahlin, CR., Young, AM., Cordier, B., Mundry, R., Wright, TF. (2014): A test of multiple hypotheses for the function of call sharing in female budgerigars, Melopsittacus undulates. Beh. Ecol. Sociobiol. 68, 145-161.
Ferreira, JCP., Fujihara, CJ., Fruhvald, E., Trevisol, E., Destro, FC., Teixeira, CR., Pantoja, JCF., Schmidt, EMS., Palme, R. (2015): Non-invasive measurement of adrenocortical activity in blue-fronted parrots (Amazona aestiva, Linnaeus, 1758). PLoS ONE 10, e0145909.
Goymann, W. (2005): Non-invasive monitoring of hormones in bird droppings: Physiological validation, sampling, extraction, sex differences, and the influence of diet on hormone metabolite levels. Annals New York Acad. Sci. 1046, 35-53.
Palme, R., Touma, C., Arias, N., Dominchin, MF., Lepschy, M. (2013): Steroid extraction: Get the best out of faecal samples. Wiener Tierärztl. Mschrift – Vet. Med. Austria 100, 238-246.
Rettenbacher, S., Möstl, E., Hackl, R., Ghareeb, K., Palme, R. (2004): Measurement of corticosterone metabolites in chicken droppings. Brit. Poultry Sci. 45, 704-711.
Sheriff, MJ., Dantzer, B., Delehanty, B., Palme, R., Boonstra, R. (2011): Measuring stress in wildlife: techniques for quantifying glucocorticoids. Oecologia 166, 869-887.
Touma, C., Palme, R. (2005): Measuring fecal glucocorticoid metabolites in mammals and birds: The importance of validation. Annals New York Acad. Sci. 1046, 54-74.
Young, AM., Hallford, DM. (2013): Validation of a fecal glucocorticoid metabolite assay to assess stress in the budgerigar (Melopsittacus undulatus). Zoo Biol. 32, 112-116.

·

Basic reporting

There are some deficiencies in this article at present:
- some terms are undefined
- the relationship between measures of cort and distress is not sufficiently discussed. What is missing is an acknowledgement that cort measures can be hard to interpret in a welfare context because the link between total cort production and the valence of experience is not simple.
- the rationale for testing effects of both rearing/current housing and season in the same study is not sufficiently explained
- the differences between the groups (WT, FP and HP is not clearly described).
- the allocation of parrots to breeding versus non-breeding season is not sufficiently well explained
- the statistical analysis is inadequately explained.
- sex is ignored in the analysis

See my detailed comments on the manuscript below.

Experimental design

The manuscript does not report a designed experiment (but this is not a criticism, merely an observation).
The design of the comparisons is poor because so many things differ between the groups compared (rearing history and current social and physical housing conditions) and this needs to be properly acknowledged and discussed.
It is unclear what the seasonal effect means for tropical animals living in the Northern hemisphere, possibly inside, in social isolation. This needs to be further clarified.

I believe that the authors do not correctly identify the design of their study and hence perform incorrect statistical analysis. Their experiment is repeated measures design (individuals were measured in two different seasons) with one between-subjects variable (WT, FP, HP).
To detect an interaction between group and season a model with an interaction term needs to be fitted to the data.

I have analysed the data provided using the correct mixed model general linear model, and I find a strong effect of group, none of season and no interaction between group and season. Therefore, I think the trend for an effect of season present in the WT group is an artefact of incorrect analysis.

Validity of the findings

The data are available as required.

The conclusions are currently not appropriately stated and are not in all cases linked to the original question set out in the introduction.

Some of the conclusions are not valid given the results obtained.

See my detailed comments below.

Additional comments

Abstract
Line 19 and 50-52: Feather picking needs to be defined better. I was not clear whether this is always self-directed.
Line 21: What is meant by “serious psychological conflict”?
Line 22: Language is not always objective (e.g. I would not describe FP as “self-defeating and self-punishing” in the abstract).
Line 22: “Ethological distress” needs defining.
Line 25: How can an animal be kept as a wild-type animal? I think wild-type is the wrong term due its usual use in genetics. Does WT mean kept in social groups or kept in large outdoor enclosures? Be more specific about what is different about the way in which the wild-type and pet parrots were kept. In the light of my subsequent reading of the paper I think it would be better to refer to your groups as human-reared and parent-reared since this seems to capture the distinction you are trying to make better.
Lines 24-27: The research question is reasonably clearly stated although please see the above comment. Does feather picking not occur in WT birds?
Line 28: It think the group of parrots studied should be described as a sample, not a group (the latter implies that they lived together).
Line 39: How can an average be a sum?
Line 42: I would not include p-values in the abstract since these are meaningless without a full description of the statistical analysis.
Lines 44-45: Concluding that cort has a role in FP seems like over-interpretation of the data: the causal arrow between cort and behaviour could be in either direction surely?

Introduction
Line 52: Is feather picking always self-directed?
Line 61: FP (a behaviour) can’t be compared to a psychological mechanism. I think what you mean here is that it has been suggested that similar psychological mechanisms might underlie both FP and trichotillomania.
Line 62: what kind of conflicts? Fights between individuals?
Line 64: you need to define ethological distress objectively.
Line 65: what is a “stress lesion”?
Line 73: If cort levels are also associated with changes in energy demand how does this relate to distress? Presumably there are reasons why animals’ energy demand may increase that are not connected to distress? The relationship between cort levels and the valence of affective state need to be more clearly discussed. A criticism of the use of cort as a welfare indicator is that it is not specific to the valence of affective states. This needs to be acknowledged and discussed.
Whilst the aims of the paper are clearly stated, it is less clear why the authors think it is interesting or important to look at both rearing history and reproductive state in the same study. The discussion gives some insight into this, but I would prefer to see the study more clearly justified in the introduction.

Methods
Line 118: it is not clear to me whether all WT parrots were both parent-reared and currently lived with exactly one conspecific partner. Were all WT parrots kept in pairs? Was the partner always of the opposite sex? How was the lack of relationship with humans established? Or is this just inferred based on how the birds were reared?
Line 122: what is meant by regular reproduction activity? How were the animals sexed?
What was the distribution of sexes in the different groups?
What determines the breeding season in parrots? How is this affected by housing in the northern hemisphere and/or in artificial light? Is the breeding season likely to be different for pet birds housed inside?
What time did the parrots wake? Peak cort is usually around awakening so this information ideally needs to be known to understand where the samples fall in relation to the diurnal cort peak. Differences in waking time could be an important source of noise in this study.
Lines 143-144: I am not clear how many days of fecal samples were collected for each parrot. Why alternate days? How many alternate days? If the criterion was 30 g material does this mean that different numbers of days were collected for different parrots? Were samples from different days pooled for each parrot prior to analysis?
Line 146: given these numbers it appears that the WT pairs of birds were pooled for cort analysis?

The ANOVA model used needs to be described properly. From the description of the experiment this appears to be a repeated measures design with one with-subjects variable (breeding season or non-breeding season) and one between-subjects variable (the three groups of birds), but I can’t tell whether this was the model fitted from the way the results are reported. Since cort levels are known to change with age it would make sense to include age as an additional covariate in the model if there is extensive variation in the age of the parrots. Why wasn’t age considered in the WT group – is it because the two individuals could differ in age?

Results
I don’t understand the statistical analysis presented in Table 1. Where do all the P-values come from? I would expect to see a single p-value for the main effect of group, a single p-value for the main effect of breeding season and a p-value for the interaction of group and breeding season. The p-values in the table suggest that a series of pair-wise tests has been performed rather than fitting a single model to the data. Or are the p-values from post-hoc tests? Either way the reporting is inadequate to understand the analysis and I suspect the statistical analysis is probably incorrect given the experimental design.
Were there any effects of sex?

Discussion
Lines 205-207: this section implies that it was an aim of the paper to establish whether FP is self-inflicted, but this was not mentioned in the introduction. I don’t understand how FP could possibly not be self-inflicted since the birds in the FP group were all solitary?

The first paragraph of the discussion should summarise the main results in relation to the aims of the paper. There is currently no mention of the aims in the first paragraph of the discussion.

Much of the following material in the discussion is actually background information that should, if anywhere, go in the introduction to the paper not the discussion. The discussion is not the place to revisit basic biology of cort.

Lines 224-227: these lines appear to provide a justification for the study that would have been better in the introduction.

Lines 229-231: Give the direction of the trend when describing it!
Lines 234-236: it is not clear to me how the result obtained allow any conclusion linking cort to experienced stress. Cort is one measure of stress, but in itself says nothing about the valence of the experience that caused cort to rise. Maybe FP birds are just more active?

Conclusion
Lines: 272-273: this is not a valid conclusion from this study because no validation of faecal cort was performed here.
Line 275: what is meant by “ethological constructs”? Say what you have actually done rather than using obscure jargon.

Lines 275-276: You should accurately describe your findings in the conclusion – give the direction of the difference.

Reviewer 3 ·

Basic reporting

Clear, unambiguous, professional English language used throughout: The manuscript needs to be revised for clarity in places. For example, line 20 incorrect plural: “Many hypothesis,” or unprofessional language (line 22 attribution of intent e.g. “self-defeating and self-punishing behaviour;” line 22 unclear terminology “ethological distress”= psychological?)

Intro & background to show context. / Literature well referenced & relevant.

Line 56: ‘is typical’ – if 90% of captive parrots don’t display behavior, and it is defined as an ‘abnormal’ behavior, then is it not by definition and atypical behavior?

Line 61-65: This needs to be developed and supported. There are more rigorous and appropriate primary source and review articles that should be considered here. Weakly referenced at the moment.

Line 87: Clarify statement ‘all companion parrots are born in captivity?’ There are many wild-caught ‘companion’ parrots, at least in some countries.

Line 87-96: this seems to ignore the not-infrequent third option of parent-rearing with human handling. How does that fit into the background of this study? Perhaps these classifications belong in the M&M, as here seem to be global statements regarding parrots in general, which is not supported.

Line 99: typo – ‘during the breeding season and in the out of’

What is the prediction for the effect of breeding season on corticosterone? Would expect it to be higher/lower, based on known literature? See Experimental Design section for further comments.


Structure conforms to PeerJ standard, discipline norm, or improved for clarity: Yes

Figures are relevant, high quality, well labelled & described. Yes

Raw data supplied (See PeerJ policy): The authors provided their original data file

Experimental design

Original primary research within Scope of the journal: Yes

Research question well defined, relevant & meaningful. It is stated how research fills an identified knowledge gap. Not explicitly. Think it is there, but need to make link between previous study and need for this study explicit in the introduction. Also and more problematic, it is unclear what hypothesis is being tested? Breeding and stress? Conspecific social isolation and stress? Mechanism of FP? Unclear how any of these are being controlled for with the current design.


Rigorous investigation performed to a high technical & ethical standard.

Was corticosterone compared between sexes? If samples are taken from bottom of cage (L145) then for paired birds, at least, could be from either/both partner. Could obscure seasonal differences in corticosterone.

Methods described with sufficient detail & information to replicate.

Methods insufficient. Authors state L104 this was a questionnaire study, but all WT birds had the same housing? Were they all from a single breeder? Limited information on birds provided in this manuscript. Perhaps in other study mentioned but lacking here.

L122 ‘all have a regular reproduction activity’ seems in opposition to L140-142, which imply breeding was occurring outside of the normal breeding season.

Validity of the findings

Impact and novelty not assessed. Negative/inconclusive results accepted. Meaningful replication encouraged where rationale & benefit to literature is clearly stated.

Data is robust, statistically sound, & controlled.

Issues here link to lack of stated hypothesis being tested. Three groups: WT control – paired and reproductively active, HP and FPP hand reared- singly housed. Unclear what is being controlled for – sexual activity, social housing?

If simply want to test for role of corticosterone in FP, then comparing the HP v. FPP seems to be more appropriate. What is the purpose of the WT group? What is being controlled for is unclear.

L224-227 seem to hint at the knowledge gap this paper is attempting to address, but it is obscured by unclear introduction/hypothesis, etc. If this was the purpose of the study need to reframe paper.

Conclusion well stated, linked to original research question & limited to supporting results.
L 205 -216 Relevance to the study presented?

L247-L262 is belied by the HP group, which does not have ‘normal or natural’ behavior and also shows no FP
L263-L264 Unclear how WT has been established to be a valid control for ‘stress coping.’ Could be suppression of HPA due to chronic stress? Experimental manipulations would be required to draw conclusions of mechanistic role.

L277 not supported by experimental design or data presented.

Speculation is welcome, but should be identified as such. Speculation not always identified.

Additional comments

Findings of a correlation between feather picking and corticosterone in pet parrots is interesting, and replicates previous studies. As written, the manuscript is not clear and many aspects of the questions being asked and/or the design are not supported. This is an important area and a worthy topic of research. A very much reduced and refined presentation of the correlational data could be useful for readers.

---

## Round 0.2 · Minor Revisions

You have completed a most thorough revision, including an improved laboratory analysis of the birds' droppings and a more appropriate statistical analysis. The manuscript is greatly improved. There are just a couple of very minor points to address before acceptance.

First, the manuscript now reads well, with the exception of the title which sounds awkward. I suggest something like this:

"An association between feather damaging behavior and corticosterone metabolite concentration in captive African grey parrots (Psittacus erithacus)"

(the detail that the CM is measured in birds' droppings is given immediately afterwards in the abstract and is not needed in the title)

In the results section please provide the descriptive data on the age of the birds separately for each of the three treatment groups, and provide an estimate of variation e.g. standard deviation.

In the discussion (e.g. line 272) there is still a slight tendency to suggest that FDB arises because of stress. As reviewer 2 pointed out previously, the direction of the causal arrow cannot be known from a cross-sectional study of this kind. As a thought experiment one could propose that some owners might use a hand-cream that irritates the skin of the birds. In this case FDB could arise as a response to skin irritation, with resultant feather loss then causing an increase in stress. This may be unlikely but it is not impossible, and longitudinal or experimental studies would be required to establish causation. This point can be made easily in just one or two sentences.

---

## Round 0.3 · accepted · Accept

I am happy to accept the manuscript but note that the Garner et al. (2008) paper clearly indicates that FDB can result from underlying skin pathologies.

I suggest the sentence starting on line 275 is re-written as:

"FDB can lead to, or result from, underlying skin pathologies that itch or irritate (Garner et al., 2008). FDB may also cause health problems related to tissue damage, hemorrhage, infection, or hypothermia (Meehan, Millam & Mench, 2003; van Zeeland et al., 2009)."

I hope you agree - if so, this can be addressed at the production stage.